# Late Pleistocene Altitudinal Segregation and Demography Define Future Climate Change Distribution of the *Peromyscus mexicanus* Species Group: Conservation Implications

**DOI:** 10.3390/ani13111753

**Published:** 2023-05-25

**Authors:** Sergio G. Pérez-Consuegra, Laura Sánchez-Tovar, Gerardo Rodríguez-Tapia, Susette Castañeda-Rico, Ella Vázquez-Domínguez

**Affiliations:** 1Departamento de Ecología, Escuela de Biología, Facultad de Ciencias Químicas y Farmacia, Universidad de San Carlos de Guatemala, Ciudad de Guatemala, Guatemala; 2Universidad Nacional Autónoma de México, Ciudad de Mexico 04510, Mexico; 3Departamento de Ecología de la Biodiversidad, Instituto de Ecología, Universidad Nacional Autónoma de Mexico, Ciudad de Mexico 04510, Mexico; santola66@hotmail.com (L.S.-T.); gerardo@iecologia.unam.mx (G.R.-T.); 4Center for Conservation Genomics, Smithsonian National Zoo and Conservation Biology Institute, Washington, DC 20008, USA; susetteazul@gmail.com; 5Smithsonian-Mason School of Conservation, Front Royal, VA 22630, USA

**Keywords:** Central America, climate change, Mexico, peromyscines, rodents, species distribution modeling

## Abstract

**Simple Summary:**

Tropical mountains are rather interesting ecosystems that exhibit a diverse array of features of an ecological niche that shapes the geographic distribution of species and their co-occurrence patterns. Both historical and contemporary factors significantly influence species and lineages diversification and distribution on mountains. We studied the *Peromyscus mexicanus* rodent group, distributed across mountains in Guatemala-Chiapas and Central America. We aimed to describe the phylogeography, demography, current distribution, and potential range changes due to future climate change. Based on a framework of genetic (mitochondrial and nuclear sequences) and ecological niche modeling methods, we show that lineages with particular ecological features and distribution on lowlands and highlands have distinctive demographic histories associated with glacial and interglacial cycles during the Pleistocene–Holocene. Additionally, the distribution range of some lineages will potentially be significantly reduced by future climate change. This information is crucial for management and conservation purposes for these lineages in particular, but also as a cautionary tale for potential climate change impacts on a variety of mountain taxa.

**Abstract:**

Mountains harbor a significant number of the World’s biodiversity, both on tropical and temperate regions. Notably, one crucial gap in conservation is the consideration of historical and contemporary patterns influencing differential distribution in small mammal mountain species and how climate change will affect their distribution and survival. The mice *Peromyscus mexicanus* species group is distributed across mountains in Guatemala-Chiapas and Central America, which experienced significant effects of glacial and interglacial cycles. We determined phylogeographic and demographic patterns of lowlands and highlands mountain lineages, revealing that the radiation of modern *P. mexicanus* lineages occurred during the Pleistocene (ca. 2.6 mya) along Nuclear Central America. In concert with climatic cycles and the distribution of habitats, lowland and highland lineages showed recent population size increase and decrease, respectively. We also estimated the current and future distribution ranges for six lineages, finding marked area size increase for two lineages for which vegetation type and distribution would facilitate migrating towards higher elevations. Contrastingly, three lineages showed range size decrease; their ecological requirements make them highly susceptible to future habitat loss. Our findings are clear evidence of the negative impacts of future climate change, while our ability to manage and conserve these vulnerable ecosystems and mountain species is contingent on our understanding of the implications of climate change on the distribution, ecology, and genetics of wildlife populations.

## 1. Introduction

A significant proportion of the World’s biodiversity is harbored along mountains, both on tropical and temperate regions. Speciation by isolation in allopatry or parapatry within montane systems has been recognized as a source of their high species richness and endemicity [1,2,3]. Integrating historical and contemporary genetic and ecological information is crucial to understand species diversity and distribution patterns among mountains and between highlands and adjacent lowlands. Furthermore, deciphering such historical and contemporary features is urgently needed to develop adequate conservation strategies in the face of climate change [4,5]. Mountains are complex ecological systems that encompass many different microclimates and niches, tightly associated with environmental gradients [6,7,8]. Elevation, slope, aspect, wind direction, rainfall, soil erosion, and overall topography are among the main features influencing biodiversity, adaptation, and speciation processes in mountains [3,9], in conjunction with latitude and global seasonal cycles [10].

Mountain systems exhibit a diverse array of features of the ecological niche that shape the geographic distribution of species and their co-occurrence patterns [9,11]. In particular, thermal niche conservatism in tropical taxa (i.e., altitudinal segregation), together with strong thermal zonation in tropical mountains, can increase allopatric isolation and act as a speciation engine [6,12]. Interestingly, mountains near the Equator, such as the central Andes in South America, exhibit less seasonality than temperate ones, due to more steady fluctuations in solar radiation and day length throughout the year, having markedly dry and rainy seasons. In Mesoamerica, a region closer to the Tropic of Cancer, mountains experience a more severe cold season, from November to March, when they are exposed to frequent cold fronts and reach extreme climatic conditions and freezing temperatures at higher summits [13,14].

Interestingly, abiotic features and species functional traits, such as body mass, vary in concert with climate on mountains, yielding different patterns of niche segregation. For instance, sister taxa of the poison frogs (genus *Oophaga*) show a parapatric distribution, replacing each other along elevational gradients [15]. Some mountain species exhibit an altitudinal version of the Bergmann’s rule—a cline of increasing body size at higher elevations (rodents, [16,17]; frogs, [18]). Additionally, lowland disturbed areas limit connectivity between populations, whereas connectivity is increased at higher elevations with more conserved forest cover [9]. It is also known that higher mountain systems show a combined pattern of elevational (parapatric) and allopatric segregation of rodent lineages (e.g., the Andes, [19]; Mexico and Central America, [17,20]).

Historical factors also significantly influence lineage diversification, rendering most mountains different and unique. In particular, Mesoamerica has rather complex phylogeographic patterns, with a mixture of organisms with North or South American affinities, as well as biota derived from local diversification processes [21,22,23]. It is during the Last Glacial Maximum and the beginning of the Holocene where a more profound footprint is observed in the distribution and demographic histories of a variety of flora and fauna in this region [22,24,25,26]. The effect of the glacial and interglacial cycles not only changed the distribution of species horizontally, but also vertically. For instance, lineage diversification at distinct altitudinal floors along mountains has been documented with paleo-pallinological data in the Andes [27] and in the Talamanca mountain range in Costa Rica [15]. Notably, one crucial gap in conservation is consideration of historical and contemporary patterns that influence the distribution in small mammal mountain species, as well as how their distribution and survival will be affected by climate change. Furthermore, understanding the impact of environmental and climatic changes on the distribution of species and communities is one crucial biological question, with urgent implications for biodiversity conservation [28,29].

The mice of the genus *Peromyscus* belong to the new world family Cricetidae, subfamily Neotominae, and tribe Reithrodontomyini [30]. Peromyscines are distributed mainly in North America (Alaska to Panama), and Mexico is considered its main diversity center [31]. Diversification of neotomines in general and *Peromyscus* in particular occurred during the late Miocene and the Pliocene [30]. At present, and based on recent revisions using molecular and morphometric analysis, the *P. mexicanus* species group is considered a monophyletic group of Pleistocene origin [31], which currently includes 12 mountain species, *P. nudipes*, *P. tropicalis*, *P. mexicanus*, *P. gymnotis*, *P. zarhynchus*, *P. gardneri*, *P. salvadorensis*, *P. nicaraguae*, *P. guatemalensis*, *P. grandis*, *P. carolpattonae,* and *P. bakeri* [17,20,32,33,34,35], distributed at lowlands and highlands across Guatemala-Chiapas and Central America mountains. 

The diversification and wide distribution of the *P. mexicanus* species group suggests they must have distinct historical (phylogeographic) and contemporary (demographic, ecological) patterns between mountains and between highlands and adjacent lowlands. In this context, the targets of the present study are to (1) determine phylogeographic patterns of mitochondrial lineages of the *Peromyscus mexicanus* group and estimate divergence times, in particular in relation to low and high mountains; (2) explore the phylogenetic relationships among lineages based on the Growth Hormone Receptor gene; (3) describe the demography of low and high mountain lineages, where our premise is that the differential effects exerted by glacial and interglacial cycles during the Pleistocene–Holocene on low and highland lineages will manifest in distinctive demographic histories; and (4) build general scenarios of distribution range changes for low and highland lineages as a result of future climate change. We demonstrate how mountain diversification generated discrete *P. mexicanus* lineages with restricted and isolated distributions, some of which will potentially be significantly reduced by future climate change. Therefore, it is crucial to take this information into account for management and conservation purposes for these lineages in particular, but also as a cautionary tale for potential climate change impacts on a variety of mountain taxa.

## 2. Materials and Methods

### 2.1. Study Site, Nuclear DNA Extraction and Sequencing

The study area encompasses the south of Mexico, including the Yucatan peninsula and Chiapas, southwards to Guatemala, Belize, El Salvador, Honduras, and Costa Rica. The mountains along southern Mexico and northern Central America (Nuclear Central America) shaped a region with a unique biota, biogeographically known as an area of endemism of the Chiapas and Guatemala highlands [36]. We here analyzed a mitochondrial cytochrome b (cyt b) dataset (1113 bp fragment) of 186 *P. mexicanus* individuals from 45 localities from southern Mexico and Central America obtained in [20], where the complete list of localities and cyt b GenBank sequences can be found.

To complement the mitochondrial results from [20], we selected a group of 20 samples that represented 15 mitochondrial distinct lineages. We amplified and sequenced an 829 bp fragment of the Growth Hormone Receptor (GHR), using primers GHR1f and GHRend1f [37]. Amplifications were performed in 25 µL reaction volume containing 50–10 ng template DNA, buffer 10×, 1 unit of Taq DNA polymerase (Vivantis Technologies, Selangor Darul Ehsan, Malaysia), 2 mM of MgCl_2_, 200 µM of each dNTP and 0.4 µM of each primer. The PCR cycling protocol was 5 min of initial denaturation at 94 °C, followed by 35 cycles of 15 s at 94 °C, 1 min at 60 °C and 1.5 min at 72 °C, and a final extension of 10 min at 72 °C. We used agarose gels (1.5%) dyed with ethidium bromide to visualize amplifications. PCR products were purified and analyzed in an ABI3730xl DNA analyzer (Applied Biosystems, Carlsbad, CA, USA) by the High Throughput Sequencing-Washington, USA. All PCR reactions included negative controls. The list of samples and GenBank accession numbers are in Appendix A.

GHR haplotypes were inferred with a maximum-likelihood method implemented in PHASE v.2.1.1 [38] and a multiple alignment was conducted with ClustalX-2 [39]. To select the best-fitted nucleotide evolution model we used jModelTest v.2.1.3 [40] and the Akaike Information Criterion (AIC). We conducted a phylogenetic Maximum Likelihood (ML) analysis with the program PhyML v.3.0 [41], using five starting “neighbor-joining” trees, with a searching option of nearest neighbor interchange (NNI), and a non-parametric branch support of approximate likelihood ratio test based on a Shimodaira–Hasewaga likelihood ratio test procedure (aLRT SH-like), with 100 replicates. A different Bayesian Inference (BI) phylogenetic analysis was completed with MrBayes v.3.1 [42], with four runs, each one with three hot and one cold Markov chains sampled every 1000 generations for 10 million generations, starting from a random tree. Convergence and stationarity were visualized with Tracer v.1.7.1 [43] with 25% generations discarded as burn-in. Topologies were visualized and edited with FigTree v.1.4.0 (http://tree.bio.ed.ac.uk/software/figtree/). We incorporated five GenBank sequences as an external group, *Calomyscus baluchi, Neotoma micropus, Isthmomys pirrensis, Reithrodontomys sumichrasti*, and *Habromys simulatus* (GQ405372.1, EF989753.1, EF989748.1, EF989824.1, KF885928.1).

### 2.2. Phylogeographic Patterns, Divergence Times and Demography

The genealogical relationships between haplotypes within the *Peromyscus mexicanus* species group were determined with a haplotype network using the 186 cyt b sequences with the program Network v.4.6.1.3 (https://www.fluxus-engineering.com/sharenet_altwin.htm), applying the median joining network algorithm. To evaluate if populations fit an isolation by distance model (gene distance with a linear and significant correlation with geographic distance), we built two matrices, one of geographic Euclidean distances estimated with the Geographic Distance Matrix Generator v.1.2.3 [44] and a genetic distance based on *Fst*. The Mantel test was performed with the IBD program [45], based on 30,000 permutations and excluding from this analysis populations with less than five sequences.

Times of divergence were estimated with BEAST v.1.7.4 [46], using the 186 mitochondrial cyt b sequences; we did not include *Calomyscus* in the external group to avoid a long branches effect. This approximation uses a relaxed phylogenetic method that is not dependent on a molecular clock. Times to most recent common ancestor (TMRCA) for main lineages were obtained using a Bayesian search with Monte Carlo Markov Chains (MCMC); we used the generalized time-reversible nucleotide evolution model with a gamma distribution and invariable sites (GTR + G + I) for all codon positions and implemented a non-correlated normal-logarithmic relaxed molecular clock and a Yule speciation process. We assumed an *a priori* constant population size. We sampled trees every 20,000 iterations for a total of 100 million generations, with 10% of initial samples discharged as burn-in. For the relaxed method, we provided calibration points and error estimations derived from log-normal distributions. We used three calibration points, the split of neotominae-peromyscines 8.6 ± 2.1 million years ago (mya) [47], the split of peromyscines 4.5 ± 1.1 mya [47], and the split of *Habromys* and *Peromyscus* 2.44 ± 0.43 mya [48]. Convergence and stationarity were visualized with Tracer.

We evaluated recent demographic histories based on the cyt b sequences of nine lineages with sample sizes of 10 individuals or more (*P. mexicanus*, denoted as lineage C in Figure 1), (*P. gymnotis*, lineage D), (*P. zarhynchus*, lineage E), (*P. zarhynchus sensu lato*, lineage G), (*P. guatemalensis*, lineage H), (*P. guatemalensis sensu lato*, lineage I), (*P. salvadorensis sensu lato*, lineage L), (*P. salvadorensis*, lineage M), and (*P. nicaraguae*, lineage N). We conducted Bayesian skyline plots analysis to examine evidence of demographic changes with BEAST, to infer potential population fluctuations over time by estimating the posterior distribution of the effective population size at specified intervals along a phylogeny [46,49]. Each of the model parameters sampled 1000 generations, with a total of 100 million generations, based also in the GTR+G+I model, under a relaxed log-normal molecular clock model, with assumption of uniform distributions and 10% discharged as burn-in. Convergence was visualized with Tracer. Lineages demographic results were complemented with their corresponding neutrality tests, including indexes of Tajima’s *D* [50], Fu and Li’s *F* and *D* [51], and Fu’s *Fs* [52], estimated with DnaSP v.5 [53].

### 2.3. Current and Future Species Distribution Models

To determine overall scenarios of potential distribution range changes as a result of future climate change, we selected six lineages from [17] that exhibit allopatric lowland and highland distributions along the Pacific and Caribbean versants, as well as distinct ecological characteristics (*P. mexicanus*, *P. guatemalensis*, *P. salvadorensis*, *P. zarhynchus*, *P. gymnotis*, and *P. nicaraguae*) (see Appendix A). For this, we first chose 46 georeferenced unique sampling localities from our six lineages, which included individuals obtained on the field, prepared as voucher specimens, and deposited in scientific collections and with corresponding geographic coordinates [17,20]. Next, we built a database with occurrences that coincided with the distribution of the selected lineages from different biological collections (Museo de Historia Natural from the Universidad de San Carlos de Guatemala, Carnegie Museum Natural History, Museo de Zoología from the Facultad de Ciencias, UNAM, and El Colegio de la Frontera Sur, San Cristóbal de las Casas, Chiapas). We removed exact coordinate duplicates to avoid pseudoreplication (Appendix A).

We used the 19 long-term average bioclimatic predictor variables from the Worldclim database, which represent a statistical summary of temperature, precipitation, and radiation [54], to perform the current distribution modeling. Climate change scenarios are not predictions of the future, but rather projections of what can happen by creating plausible descriptions of possible climate change [55]. As such, we obtained the digital surfaces from the CGIAR Research Program on Climate Change, Agriculture and Food Security (CCAFS9 (http://www.ccafs-climate.org/data) from two scenarios, SREASA2 and SRESB1, which consider medium-high and low-medium emissions, respectively. We used the general circulation model cccma_cgcm3_1_t47 for three time periods (2020, 2050, 2080) with a 30″ arc spatial resolution (~1 km^2^). We thus had seven final climatic files, one for the current data, plus six including the two scenarios per time period (one each for A2 and B1; 2020, 2050, and 2080).

To adequately select the area for modeling we considered the different lineages’ distributions, based on which we built four envelopes with ArcMap 9.2 (Appendix A), taking into account the proximity of the occurrence records and the known geographical distribution of each lineage [17,20], grouped as follows, Mexicanus (*P. mexicanus*), Gymnotis (*P. gymnotis*), Nicaraguae (*P. nicaraguae*), and Others (*P. guatemalensis*, *P. salvadorensis*, *P. zarhynchus*). We created models using the algorithm Maxent 3.2 [56], which considers the probability distribution of maximum entropy subject to constraints imposed by a known distribution of species (i.e., lineages) and by the environmental conditions across the study area [57]. The extrapolation of these values to a given area results in a probability distribution map ranging from 0 to 1. Maxent has proven to generate good results even with small sample sizes (<10) [56,58]. The modeling parameters for Maxent were set to exclude extrapolation and clamping, using a logistic output format, bootstrapping (1000 iterations), 10 replicates, and a 10-percentile threshold; 80% of the occurrences were randomly selected and used to train the models and 20% to validate them. Runs were performed for the seven scenarios for each of the six lineages. The resulting models were converted to raster average maps, which were next reclassified based on the equal training sensitivity and specificity (using ArcView 3.1 and ArcMap 9.2). Finally, we performed ‘general combine’ with ArcView 3.1 to combine the current distribution maps with each of the six future scenarios per lineage; the resulting maps show the projected ranges plus the intersection areas between the current and future ranges, thus pixels outside the intersection can be visualized. We report the value of the area under curve (AUC), which ranges from 0 to 1; a value of 0.5 indicates that the model performance is not better than random, while values closer to 1 indicate better performance of the model.

## 3. Results

### 3.1. Phylogeography and Demographic Patterns

For the nuclear gene (Growth Hormone Receptor) we obtained 826 bp sequences for 18 individuals and 12 lineages (lineages A, E, and L did not amplify despite multiple trials), which included 812 invariable sites, 14 polymorphic sites, and 10 parsimony informative sites. We identified 14 different haplotypes (*h* = 0.8111), low nucleotide diversity (*π* = 0.00218) and a mean number of nucleotide differences between haplotypes (*k*) of 1.802. Neutrality tests showed that data fit the hypothesis of neutrality (Tajima’s *D* = −1.501, *p* > 0.1; Fu and Li’s *F* = −0.754, *p* > 0.1; Fu’s *F_S_* = −8.217, *p* > 0.1). The substitution nucleotide model selected was HKY+G, with the following parameters, frequencies of bases A = 0, 2947, C = 0.2724, G = 0.2205, T = 0.2125; nst = 2; rates = gamma; and α = 0.3680.

The nuclear phylogenetic results showed a similar topology for both ML and BI analyses, depicting three main clades with high support values (pp = 0.9–1.0), yet where haplotypes are not well differentiated (Appendix A). Despite the fact that nuclear phylogeny had a shallow resolution, it is in agreement with the recognized mitochondrial monophyly of the *P. mexicanus* group (see Figure 4 in [17]). The Mantel test results did not support an isolation by distance pattern (r = −0.025, *p* = 0.40).

The haplotype relationships indicated by the haplotype network results showed high correspondence with the lineages’ geographic (Appendix A) and elevation distribution, denoting five main branches joined by a central group of hypothetic haplotypes. Following the same lineages classification (with letters) as in Pérez-Consuegra and Vázquez-Domínguez [17], the branches are distributed as follows (Figure 1a,b and Appendix A), lineage B (*P. tropicalis*) in the low altitude mountains of north-eastern Guatemala (Cerro San Gil, Mayan Block); lineages E, F, and G (*P. zarhynchus sensu lato*; *P. gardneri*) in the mid and high mountains of the Altos de Chiapas and central Guatemala mountains north of the Polochic Fault (Mayan Block); a third branch shares lineages in the Mayan Block (H and K) and the Chortis Block (lineages I, J, L, M) and a fourth in the Chortis Block (lineage N, *P. nicaraguae*) and the Motagua-Polochic Fault System (lineage O); lineage A (*P. nudipes*) from the highlands of the Sierra de Talamanca in Costa Rica; lineage C (*P. mexicanus*) on the mountains of central México (Veracruz to Hidalgo) north of the Isthmus of Tehuantepec; and lineage D (*P. gymnotis*), distributed in lowlands around the Isthmus of Tehuantepec and the Pacific slope of Chiapas and western Guatemala.

Divergence times estimates dated the time to the most common recent ancestor (TMCRA) of the split between the *Peromyscus mexicanus* species group from *Habromys* (node 1 in Figure 1b) approximately 4.22 (95% HPD: 3.43–5.17) million years ago (mya), during the mid-Pliocene. The divergence between lineages I, II, and III from Clade IV, the most diverged of all at 2.62 mya (2.52–2.72; Node 2) during the early Pleistocene. The diversification of Clade IV initiated with the split of lineages E, F, and G (Mayan Block) from lineages H, I, J, K, L, M, N, and O (Chortis Block and some from the Mayan Block) during early Pleistocene (Node 3; 2.38 mya, 2.05–2.62). It was followed by two split events that separated Clade II (Motagua-Polochic-Jocotán fault system in Guatemala) from Clades I/III, and a bit later, the split of Clade I from Clade III, around 2.2 (Node 4; 1.6–2.6) and 1.94 (Node 5: 0.74–2.0) mya, respectively. An additional split event took place around 1.96 mya (Node 6; 1.56–2.32), involving various lineages of mid elevation in central Guatemala, Honduras, and Nicaragua (lineages N and O), followed by the differentiation of high elevation lineages in the Maya Block and the volcanic chain in Guatemala (H, I, J, K; 1.3 mya, 1.2–2.0; Node 9) from those of mid-elevations of the Chortis Block and west side of the Honduras Depression (M, L; 1.18 mya, 0.7–1.6; Node 10).

Skyline plots showed an overall historical demographic stability across lineages, in conjunction with patterns of recent population expansion or contraction (Figure 1c). Middle and high elevation lineages (C, E, G, and L) showed a signal of population size reduction during the Holocene, beginning after the Last Glacial Maximum (ca. 25 ky ago). An exception is lineage I, from high elevations along western Guatemala and Chiapas (Triunfo), which showed an older population expansion beginning approximately 50 ky ago. On the contrary, low elevation lineages (D, N, and M) showed a pattern of population expansion after the Last Glacial Maximum. Lineage H, which occupies the highest elevations at the Sierra de los Cuchumatanes (3000 m, north of the Polochic-Motagua fault zone), did not show a clear pattern and instead its population size remained mostly stable, with a sign of very recent contraction. 

### 3.2. Niche Modeling 

The distribution model with the best performance for each lineage per time (current, 2020, 2050, and 2080) showed area under the curve (AUC) values above 0.97, except those for *P. gymnotis* that ranged from 0.768 to 0.819 (Appendix A). The current potential distribution areas estimated ranged from 3095 to 59,521 km^2^ (Appendix A). Regarding the future potential distribution, total areas differed depending on the IPCC scenarios modeled, ranging from 2285 to 52,669 km^2^ (Appendix A). Based on these projections, we estimated the current and future potential area change (increase or reduction) per lineage for 2020, 2050, and 2080, respectively (Table 1; Figure 2, Figure 3 and Figure 4). Results indicated scenarios where lineages having both the most extensive and the smallest current distribution areas are consistently (both scenarios and across years) likely to suffer significant decrease in distribution area, as shown by *Peromyscus nicaraguae*, *P. guatemalensis*, *P. gymnotis*, and *P. zarhynchus* (Figure 2 and Figure 3). Only *P. mexicanus* and *P. salvadorensis* exhibit potential area increases (Figure 4).

## 4. Discussion

### 4.1. Phylogeographic and Demographic Patterns of Lowland and Highland Mountain Lineages

The distribution of species through space and time is driven by a dynamic interaction between ecological and evolutionary processes. Ecological interactions can, for instance, dampen or promote evolution through their influence on selection regimes, as well as population size and connectivity [3,59,60]. Concomitantly, evolutionary processes can affect ecological population and community dynamics through genotype and phenotype environment feedbacks [15,61,62].

The estimated divergence times show that the ancestor of the *P. mexicanus* group inhabited the region approximately 4 million years ago (mya), during the so called ‘Middle Pliocene Thermal Optimum’, when the earth experienced a long period of relatively warm and tropical climates that dominated southern North America [63]. During this period, sea level was higher, likely isolating Nuclear Central America via shallow sea channels along the Isthmus of Tehuantepec and the Nicaraguan Depression, acting as biogeographic barriers [21,64,65]. The radiation of modern lineages occurred later, during the Pleistocene (ca. 2.6 mya). The haplotype network suggests that this mitochondrial ancestor inhabited Nuclear Central America, a region that harbors both the majority of the lineage’s ancestors and the highest diversity of the group. The phylogeny obtained with the nuclear marker showed low resolution in comparison to the mitochondrial one, not an unexpected result given the fact that the diversification of the *P. mexicanus* lineages is relatively recent. High Pleistocene–Holocene diversification is a pattern commonly seen in species that diversified recently along Nuclear Central America (e.g., [21,66]).

One of the most significant effects of glacial cycles in Central America mountain systems was the drying of lowland areas and valleys, while also markedly cooling the highlands, while the contrary happened during the interglacial phases [67,68]. Through these cycles, cold humid areas occupied extensive areas together with reduced tropical dry lowlands during the Last Glacial Maximum (LGM), which changed to decreased cold habitats on mountain peaks and wide tropical habitats extending from mid to low elevations during the Holocene (Appendix A) [14,69]. Such temperature and humidity oscillations had cascading effects on vegetation and habitat features, which changed altitudinally along mountains, causing concomitant distribution changes of mountain fauna [11,27,70]. For example, the viper snake *Montivipera raddei* expanded its distribution range altitudinally during the LGM, following its optimal habitat that rendered a fragmented allopatric distribution currently restricted to mountaintops [71], a pattern also documented for other taxa-like birds [72] and trees [73], to mention a few. The *P. mexicanus* group was not an exception, as shown by the distinct phylogeographic lineages and the recent demographic changes of lowland and highland lineages.

Our demographic history results of nine different elevation lineages reveal they experienced distinctive demographic patterns. Low (lineage D) and midland (M, N) lineages showed a signal of recent increases in population size, while population sizes of highland lineages (E, G, H) decreased. The exception was midland lineage C (*P. mexicanus*), showing a marked population reduction; this is likely the result of *P. mexicanus* having a widespread distribution and ample elevational range. Moreover, our findings suggest that colonization of highlands from lowlands, or vice versa, occurred multiple times during the Pleistocene. An example is the colonization of the highlands of Sierra de Talamanca in Costa Rica by lineage A (*P. nudipes*), and that of lineage C from the lowlands to mid-elevation areas of the mountains in central Mexico. The generalized increase in temperature and humidity in Central America during the Holocene [15,69] favored the expansion of lowland mountain forest to high elevations, with more humid lowland areas. Lineage H from Cuchumatanes and lineage I from Sierra Madre occupy the highest altitudes (2500 to 3400 m) on different mountains. The fact that they are sister lineages with relatively low genetic distances [20] suggests they had contact likely through habitat corridors that intermittently opened and closed between mountains, and that they diversified in the recent past. Hence, our findings show that particular ecological requirements and dispersal abilities of lineages significantly influenced the genetic and phylogeographic patterns observed, yielding specialized climatic niches and altitudinal stratification [3].

### 4.2. Overall Distribution Range Changes Associated with Future Climate Change

The potential distribution obtained for each of the six *Peromyscus mexicanus* lineages is a geographic representation of their environmental niche. We acknowledge that future models are an approximation based on which potential distribution change scenarios can be described, but which are not explanations nor accurate predictions; also, we acknowledge that many future modeling algorithms have been proposed [74,75]. We chose to work with two general scenarios that would allow us to fulfill our objective of analyzing general potential range size change, using a broad-spectrum considering medium-high and low-medium emissions (SREASA2 and SRESB1, respectively). Therefore, we only modeled scenarios for the overall potential future distribution of each lineage, based on climatic variables jointly with three future time periods (general circulation models). Interestingly, this approach allowed us to describe changes in range size (area) but also to identify spatial distribution changes that did not necessarily imply a size modification.

Furthermore, the variability of the current and future potential range size estimations of the six lineages revealed the ecological complexity of this mice group. Our findings show that the future scenarios of the distribution area size increased considerably for two lineages, irrespective of the emissions considered, *P. mexicanus* and *P. salvadorensis*. These lineages have, comparatively with the other four, small (8246 km^2^) and medium (17,613 km^2^) current areas, thus size does not seem to be a factor determining the patterns of future change. What distinguishes these lineages is that the vegetation type and elevation along their distribution facilitate migrating towards higher elevations. *Peromyscus mexicanus* is distributed from 500 to 1700 m along the Sierra Madre Oriental and Sierra de los Tuxtlas in Mexico; the future scenario of its distribution follows a southeastern direction, coinciding with the presence of low- to mid-elevation cloud forest. Comparatively, *P. salvadorensis* reaches a much higher elevation, from sea level to 2500 m, which is likely associated with the fact that no fragmentation or habitat loss was observed. We suggest that the increase in area result is because this lineage could potentially extend its distribution along the plains and dry valleys of tropical deciduous forest dominating its current habitat.

In contrast, three lineages showed various degrees of future range size decrease, from small (20–30%, scenario A2 in *P. guatemalensis*) to moderate (30–50%; *P. gymnotis*) and severe (60–80%; *P. nicaraguae*). *Peromyscus guatemalensis* has the current most restricted distribution, associated with broad leaf evergreen forests on cold highlands between 1700 and 3200 m, which represent key factors of this lineage’s ecology. Both future scenarios for *P. guatemalensis* agree with a distribution restricted to the mountains north of Chiapas and northwest of Guatemala. Thus, a future change in temperature and humidity will cause significant habitat loss in these mountains, combined with the impossibility of migrating to higher elevations [76]. It is important to highlight that part of the predicted future range for this particular lineage includes protected areas for conservation.

The scenarios for *P. gymnotis* are highly concordant with the vegetation types it inhabits, broad leaf evergreen forests, tropical montane humid forests, and mid-elevation cloud forests. Future habitat loss for this lineage follows the Pacific coast of Chiapas (Sierra Madre del Sur) to Guatemala in a southeastern direction. Contrastingly, the distribution change for *P. nicaraguae* does not follow a specific pattern, resulting in multiple fragmented areas. Notably, *P. gymnotis* and *P. nicaraguae* share the characteristic of a coastal distribution, from sea level to 1700 and 2000 m, respectively, dominated by evergreen vegetation and extreme precipitation regimes. In addition, they have the most extensive current distribution (51,063 and 59,521 km^2^, respectively). Such ecological settings make these lineages highly susceptible to future habitat loss, since they cannot migrate or extend their coastal limited distribution, in addition to the vegetation strictly dependent on high precipitation levels. On the other hand, these lineages could also already occupy most of their suitable habitat, hence the result of high range size decrease.

The potential distribution ranges identified for these lineages could encompass, currently or in the future, areas that have different degrees of perturbation, namely where land use change, deforestation, or fragmentation has happened or will occur in the near future, rendering them unsuitable. This is particularly important for lineages that would reduce their distribution range with climate change, since they will face a more severe impact [77]. Additionally, even those that could increase their ranges will be negatively impacted, since the potential area could actually be significantly less. Hence, it would be important to explore these areas in the field, to evaluate their condition and viability and to develop restoration programs for those that will be unsuitable in the future, for these lineages and undoubtedly for other taxa.

## 5. Conclusions

The phylogeographic dynamics of species and lineages are shaped by historical as well as ecological boundaries, as is the case of the *Peromyscus mexicanus* group. Indeed, considering the historical patterns determining diversification and distribution was crucial to delineate evolutionary distinctive lineages on these mountain systems, which enable us to evaluate contemporary patterns of distribution and potential climate change effects on the lineages’ range size and survival [5]. Notably, mice of this group have been effective in sorting biogeographic barriers and Pleistocene–Holocene climate cycles, dispersing altitudinally, adapting allopatrically to the lowlands and highlands in intimate association with the environment. Nonetheless, as we demonstrate, these lineages are highly susceptible to modern accelerated climate changes that are not equivalent to the historical scale of their evolutionary history [28]. 

Our findings, although based on a particular group of mountain mice, are clear evidence of the negative impacts of future climate change on biodiversity. The ecosystems where these lineages are distributed are severely threatened by changes in temperature and precipitation. Moreover, anthropogenic activities and high deforestation rates exert strong pressures on them, fragmenting current habitats and shrinking potential future suitable areas. All these factors can have further consequences, reducing genetic variability and evolutionary potential of individuals, limiting their capacity to cope and adapt to environmental changes. We thus emphasize the increasing urgency of implementing actions to control anthropogenic stressors and to mitigate the effects of climate change in these extraordinary mountain systems, while focusing on increasing the number and extension of natural reserves and protected areas. Undoubtedly, the more we know and understand of the implications of climate change on the distribution, ecology and genetics of wildlife populations, the better strategies we can develop to preserve biodiversity, at present and for the future.

## Figures and Tables

**Figure 1 animals-13-01753-f001:**
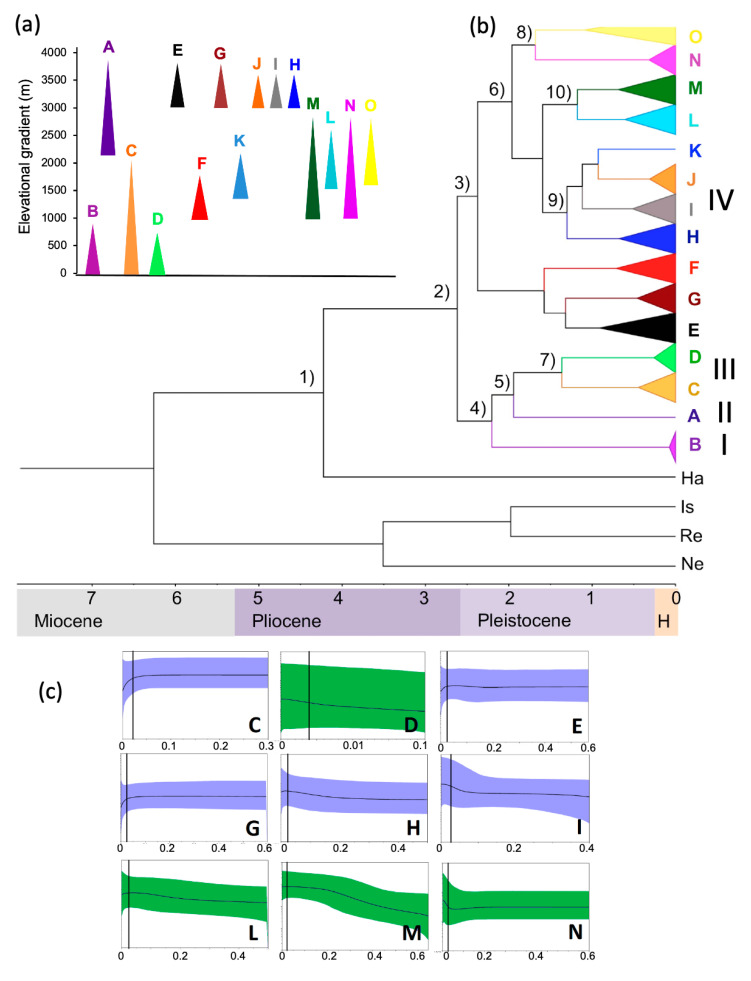
(**a**) Elevational gradient occupied by the 15 lineages of the *Peromyscus mexicanus* species group, where the allopatric and sympatric distribution across mountains is shown. (**b**) Divergence-time estimation (time-scale in millions of years; mya) of the 15 lineages (letters A to O) and clades (I–IV) indicated as in Pérez-Consuegra and Vázquez-Domínguez [17]; *Habromys* (Ha), *Isthmomys* (Is), *Reithrodontomys* (Re), and *Neotoma* (Ne) used as outgroups. Nodes 1–10 correspond to the estimated times as explained in the main text. (**c**) Demographic results based on Skyline plots for nine lineages; green and purple colors indicate the low- and mid-land (0–2500 m) and the high-land (reaching above 3000 m) lineages, respectively. The black vertical line within each graphic depicts a reference to the time of the Last Glacial Maximum, ca. 20,000–25,000 years ago. The x-axis denotes time in millions of years ago (mya), and the y-axis the estimated effective population size on a logarithmic scale (values not shown).

**Figure 2 animals-13-01753-f002:**
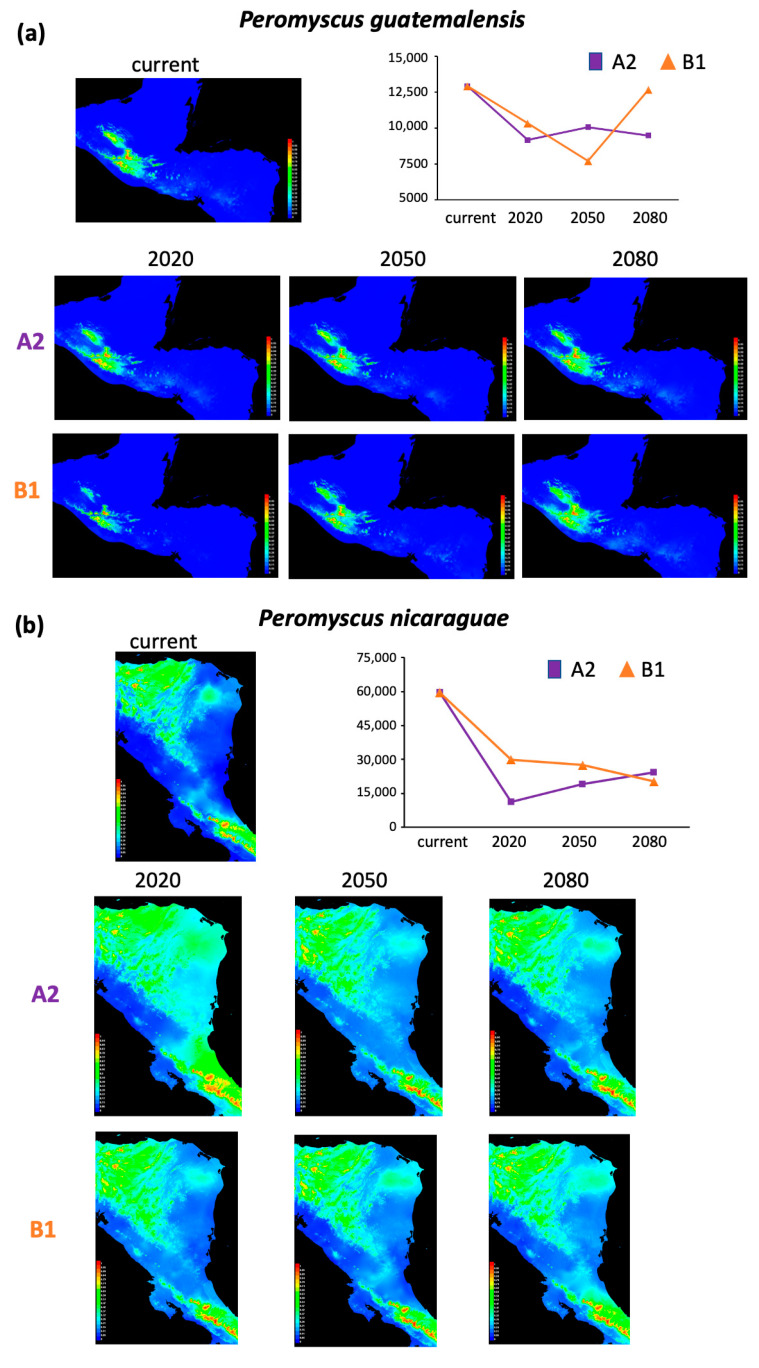
Current and future potential distribution areas (km^2^) of (**a**) *Peromyscus guatemalensis* and (**b**) *P. nicaraguae*, based on two scenarios, SREASA2 and SRESB1, that consider medium-high and low-medium emissions, respectively, and the general circulation model cccma_cgcm3_1_t47 for three time periods (2020, 2050, and 2080).

**Figure 3 animals-13-01753-f003:**
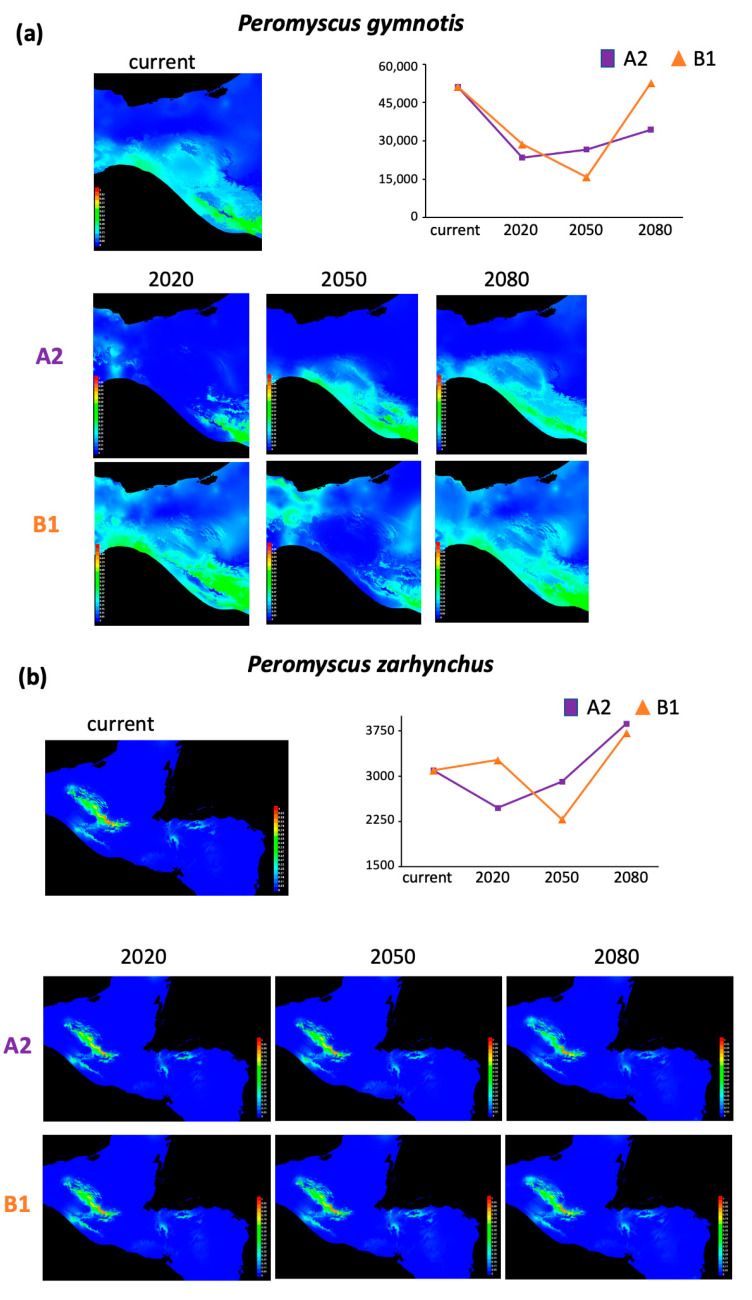
(**a**) Current and future potential distribution areas (km^2^) of (**a**) *Peromyscus gymnotis* and (**b**) *P. zarhynchus*, based on two scenarios, SREASA2 and SRESB1, that consider medium-high and low-medium emissions, respectively, and the general circulation model cccma_cgcm3_1_t47 for three time periods (2020, 2050, and 2080).

**Figure 4 animals-13-01753-f004:**
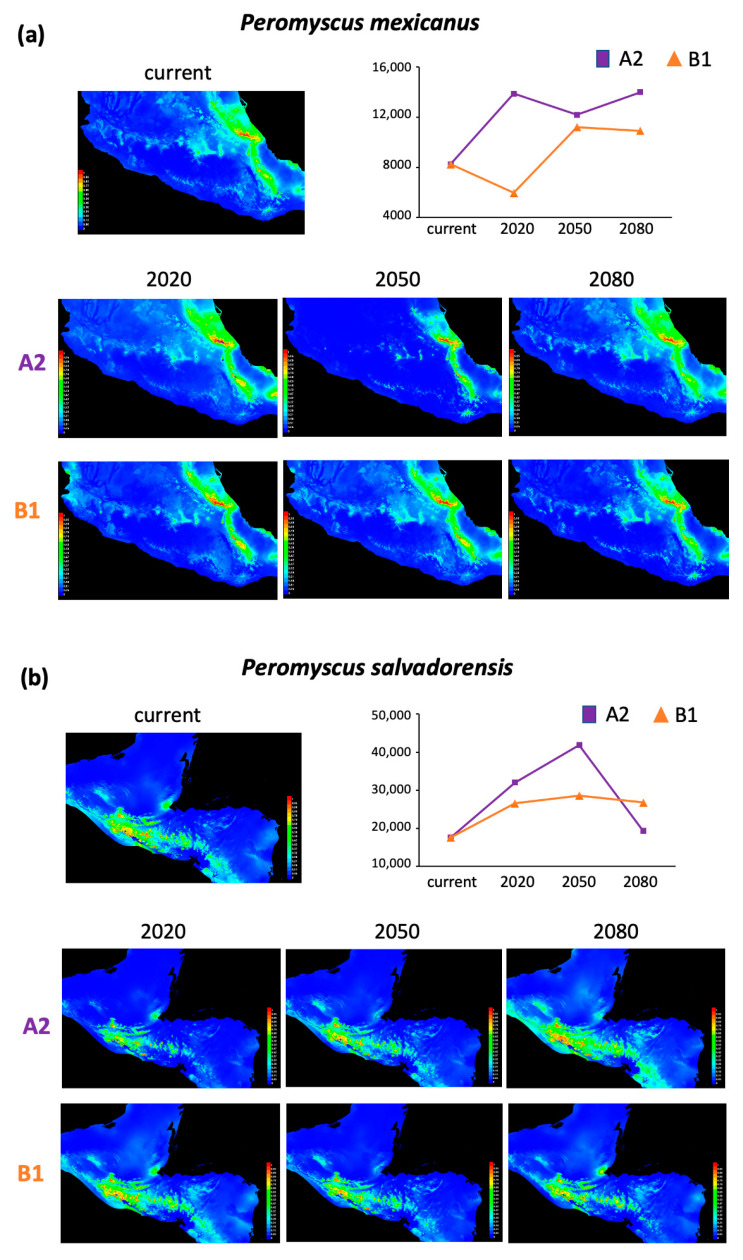
(**a**) Current and future potential distribution areas (km^2^) of (**a**) *Peromyscus mexicanus* and (**b**) *P. salvadorensis*, based on two scenarios, SREASA2 and SRESB1, that consider medium-high and low-medium emissions, respectively, and the general circulation model cccma_cgcm3_1_t47 for three time periods (2020, 2050, and 2080).

**Table 1 animals-13-01753-t001:** Overall potential future distribution range changes (in km^2^), in relation with the estimated current distribution areas, and percentage of loss or gain for six lineages of the *Peromyscus mexicanus* group. Future projections were completed based on scenario A2 (SREASA2) and scenario B1 (SRESB1), which consider medium-high and low-medium emissions, respectively, and using the general circulation model cccma_cgcm3_1_t47 for three time periods (2020, 2050, and 2080). Numbers in bold indicate cases where areas decreased.

Lineage		Scenario A2			Scenario B1		
	Current	2020	2050	2080	2020	2050	2080
*P. mexicanus*	8246	5620	3939	5738	**2298**	2956	2656
		+68%	+48%	+70%	**−28%**	+36%	+32%
*P. gymnotis*	51,063	**27,728**	**24,478**	**16,750**	**22,391**	**35,263**	1606
		**−54%**	**−48%**	**−33%**	**−44%**	**−69%**	+3%
*P. nicaraguae*	59,521	**48,216**	**40,455**	**35,165**	**29,586**	**31,921**	**39,207**
		**−81%**	**−68%**	**−59%**	**−50%**	**−54%**	**−66%**
*P. guatemalensis*	12,918	**3759**	**2861**	**3439**	**2598**	**5220**	**261**
		**−29%**	**−22%**	**−27%**	**−20%**	**−40%**	**−2%**
*P. salvadorensis*	17,613	14,424	24,285	1705	8.965	10,995	9178
		+82%	+138%	+10%	+51%	+62%	+52%
*P. zarhynchus*	3095	**622**	**184**	775	174	**809**	620
		**−20%**	**−6%**	+25%	+6%	**−26%**	+20%

## Data Availability

The GenBank Accession numbers for the GHR gene sequences obtained in this study are OQ992674–OQ992687, also included in the Appendix A.

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
