# Peer review of "Late Pleistocene Altitudinal Segregation and Demography Define Future Climate Change Distribution of the *Peromyscus mexicanus* Species Group: Conservation Implications"

_animals, 2023, doi:10.3390/ani13111753_

Round 1

Reviewer 1 Report

The manuscript aimed to report the phylogeography, demographic history, current distribution and potential range changes due to future climate change of the Peromyscus mexicanus species group in Nuclear Central America. The authors stated that they combined genetic (mtDNA and nuDNA) data and ecological niche modeling. The mtDNA cytb data set were retrieved from published work of Pérez-Consuegra and Vázquez-Domínguez (2015). However, the novel nuclear data are insufficient to test the phylogeographic hypothesis inferred from the mtDNA cytb data set in Pérez-Consuegra and Vázquez-Domínguez (2015). One of my major queries, which I hope effective, regards the divergence dating which has an effect on the paper and the hypothesis in the discussion. The authors did not set a reasonable calibration point with respect to the split of Habromys and Peromyscus based on León-Paniagua et al. (2007). Another concern regards Bayesian skyline plots analysis. To attach a real time-scale to past demographic events, the authors should pay attention to selection of age calibrations. However, it is not clear whether rate (cytb) or time duration was used in the present study. I thus believe the manuscript is not suitable for publication in the journal in its present form.

 Overall, this manuscript suffers from some issues in terms of text and displays. Below are some detailed comments and suggestions that if addressed by the authors they would potentially improve the current version of the manuscript.

Author Response

Response submitted as a pdf 

Reviewer 2 Report

Dear Authors, 

Thank you very much for such an excellent effort. I see your point. However, I would like to recommend some recommendations. 

Line 29, delete one point at the end of the sentence.

Line 51, the introduction initiates in the same way that the abstract. I would recommend changing the sentence to make your paper more dynamic.

Line 94 is another paragraph that mentions the same as the abstract. I would recommend organizing better this part of the document.

Line 180 Monte Carlo Markov Chains (MCMC), please check.

Line 270 Is there another paper(s) that supports your assumptions to avoid self-references? (same in line 275 and line 291).

I agree with your discussion that the severe changes limited the adaptation of Peromyscus (and other species). 

Author Response

Dear Authors, 

Thank you very much for such an excellent effort. I see your point. However, I would like to recommend some recommendations. 

Response: We deeply thank the reiewer for the time and comments provided.

Line 29, delete one point at the end of the sentence.

Response: Done

Line 51, the introduction initiates in the same way that the abstract. I would recommend changing the sentence to make your paper more dynamic.

Response: The reviewer is right. We modified the sentence in the introduction, expressing the original idea.

Line 94 is another paragraph that mentions the same as the abstract. I would recommend organizing better this part of the document.

Response: We also modified the sentence in the introduction, expressing the original idea.

Line 180 Monte Carlo Markov Chains (MCMC), please check.

Response: Corrected

Line 270 Is there another paper(s) that supports your assumptions to avoid self-references? (same in line 275 and line 291).

Response: Throughout the manuscript we include as varied and complementary references as possible; along the lines the reviewer indicates, we are specifically refering to the information and results obtained in one of our previous studies (Pérez-Consuegra and Vázquez-Domínguez 2017) thus we could not change the reference.

I agree with your discussion that the severe changes limited the adaptation of Peromyscus (and other species).

Response: Thanks

Round 2

Reviewer 1 Report

After a careful reading, I found that some major comments have not been fully considered. The manuscript has not been sufficiently improved to warrant publication in Animals.

The authors did not respond to one of my concerns that the nuclear data is insufficient to test the phylogeographic hypothesis inferred from the cytb data in Pérez-Consuegra and Vázquez-Domínguez (2015). Another regards Bayesian skyline plots analysis. To attach a real time-scale to past demographic events, rate or time duration should be selected as age calibrations. However, the authors did not state it at all. See lines 226-233.

Discussion

Lines 428-431

This sentence is problematic and will be misleading with regard to phylogenetic relationships in the tribe Reithrodontomyini. Firstly, it is evident that there is not any novel finding on phylogeney in the present study. The sample for the nuclear data is too small, and not sufficient to test the phylogeographic pattern inferred from the mtDNA cytb in Pérez-Consuegra and Vázquez-Domínguez (2015). Secondly, the intergeneric relationships of Reithrodontomyini were not clear, and the sister group of genus Peromyscus is still equivocal, and the intra-relationships of Peromyscus are unresolved yet. As demonstrated in previous studies, the sister taxon of the Peromyscus mexicanus species group was unclear either.

Lines 704-705: Pay attention to the format.

Musser, G.G.; Carleton, M.D. Superfamily Muroidea. 2005. In Mammal Species of the World: A Taxonomic and Geographic Reference. 3rd edition. Wilson D.E.; Reeder D.M. Eds.; Johns Hopkins University Press, Baltimore, Maryland, USA; 2005, 894-1531.

There are still some small typos and other errors. A thorough language check is necessary.

For example, line 168:

“The list of samples and GenBank accession numbers are in Table S1.”

In the supplementary material

Figure S2, not 2018, but 2017.  Please revise the citation and follow the rule of Animals.

Figure S3: the citation has not followed the rule of Animals.

ca. ca.

Author Response

Response in a pdf file
